# Engaging Occupational Safety and Health Professionals in Bridging Research and Practice: Evaluation of a Participatory Workshop Program in the Danish Construction Industry

**DOI:** 10.3390/ijerph18168498

**Published:** 2021-08-11

**Authors:** Mikkel Brandt, Ninna Maria Wilstrup, Markus D. Jakobsen, Dwayne Van Eerd, Lars L. Andersen, Jeppe Z. N. Ajslev

**Affiliations:** 1National Research Centre for the Working Environment, Department of Musculoskeletal Disorders and Physical Workload, DK-2100 Copenhagen, Denmark; nmw@nfa.dk (N.M.W.); mdj@nfa.dk (M.D.J.); lla@nfa.dk (L.L.A.); jza@nfa.dk (J.Z.N.A.); 2Institute for Work & Health, Toronto, ON M5G 1S5, Canada; dvaneerd@iwh.on.ca; 3Sport Sciences—Performance and Technology, Department of Health Science and Technology, Aalborg University, DK-9100 Aalborg, Denmark

**Keywords:** musculoskeletal disorders, construction sector, blue collar workers, occupational risk prevention, construction workers

## Abstract

Engaging occupational safety and health (OSH) professionals has scarcely been evaluated as a means for transferring knowledge to practice about physical workload in the construction industry. The aim of this work was to examine how participants used and incorporate research-based knowledge from a three-day training course into practice. Twenty OSH professionals from the Danish construction industry participated in a workshop-training course. Researchers presented new knowledge and results about physically demanding work. The participants selected which themes they wanted to work with and developed an action plan. Evaluation was done using surveys and phone interviews. Analysis was based on how the OSH-professionals describe themselves, organizations, and the construction industry. Participant’s average scores on the level of implementation of their chosen action plans were 3 (on a response scale from 1–5, where 1 is ‘to a very low degree’ and 5 is ‘to a very high degree’) immediately after the workshop program and 2.5 at follow-up. Qualitative evaluations showed that actions had been initiated, and some progress had been made. The participants were satisfied with the workshop course and the possibility to increase their knowledge through inputs from researchers and colleges and strongly believe that they would succeed with implementing their action plans in the future.

## 1. Introduction

Musculoskeletal disorders (MSD) are still a problem across different industrial sectors. The Danish Work Environment and Health study shows that risk factors for MSD and loss of ability to work, related to physical strain at work, are a major problem in the working environment in Denmark [1]. The same pattern can be seen across different European countries [2]. Prevention of musculoskeletal pain should be a high priority, as pain will often recur once it has been present [3,4,5].

In specific industries, physical workload plays a particularly important role in the development of MSD. In particular, the construction industry is characterized by many repeated manual lifts, including heavy lifts, and work with bends and twists in the back [1]. These high physical work demands increase the risk of sickness absence and disability pension [6,7,8,9]. A recent study based on a registered follow-up of the Danish Work Environment and Health study shows that physical loads on the back—forward bending, bending, and twisting in the back as well as lifting, carrying, pulling, and pushing loads—are associated with increased risk of long-term sickness absence [9]. Concrete workers and bricklayers are, to a great extent, exposed to these risk factors [1]. Therefore, employees and managers in the construction industry often seek practical and helpful knowledge to reduce hazards related to physically demanding work tasks [10].

For workplaces to organize work safely, task-specific knowledge of physical work demands is beneficial, as this can allow for more targeted preventive efforts. With the technological development of recent years, physical workloads can be determined directly at the workplace during the working day [11,12]. In a recent study, we used technical measurements, i.e., electromyography (EMG), motion sensors, and video data, in a participative intervention with concrete workers and bricklayers [13,14]. Study findings confirmed that, although there is already general knowledge about physical exertion (e.g., heavy lifting) and the risk of MSD, this knowledge can be challenging to implement in the construction industry. For workplace interventions, many researchers emphasize the active involvement of employees and managers as central to the successful implementation and maintenance of the intervention [15,16,17,18]. However, there is a need for more practically applicable and usable knowledge that makes sense for both employees and managers, including occupational safety and health (OSH) professionals, who have a vital role in injury prevention.

The focus on evidence-based practice in the work environment is increasing but translating scientific evidence-based knowledge into practice can be challenging [19,20,21]. However, engaging OSH professionals, e.g., OSH managers and OSH coordinators, have scarcely been evaluated as a means for transferring knowledge to practice in the construction industry. This knowledge-transfer issue can be based on several different conditions, e.g., understanding and agreement on the problem [15] or prioritization and implementation of actions at the workplace [22]. Previous research shows that one of the essential elements in ensuring a good intervention in the work environment is that the development, adaption, and decision-making regarding the specific concern occurs in close collaboration between researchers and the users or knowledge brokers [23]. In response to this, we invited OSH professionals from the construction industry to participate in an action-oriented and participatory course, with workshops on preventing physical exertion among construction workers; one that prompted the OSH professionals to discuss, collaborate on, define, and implement measures for preventing physical exertion in their respective work settings. The present project aimed to examine how the participants use and incorporate research-based knowledge from a three-day training course in practice.

## 2. Materials and Methods

### 2.1. Participants

Twenty OSH professional were recruited from companies in the Danish construction industry, that were working throughout the country. The OSH professionals were all Danish and came from companies that represent concrete workers, bricklayers, carpenters, and OSH consultants, working across the construction sector, and all represented blue-collar workers. The inclusion criteria were that the work-environment professionals represented companies with >30 employees and must have be in a position to influence the work environment in the company. A company’s size is of crucial importance to be able to allocate resources to participate in the development of work environment initiatives [24]. Furthermore, we aimed to recruit broadly within the industry, both in terms of companies and geographical distribution.

### 2.2. Ethics

The study was notified to and registered by the Danish Data Protection Agency (Datatilsynet; journal number 2015-57-0074). All participants signed informed consent before the workshop course. All data were processed and analyzed anonymously.

### 2.3. Workshop Course

The work-environment professionals participated in three workshops, each lasted 4 h, at five-week intervals, held at the National Research Centre for the Working Environment, Copenhagen, Denmark, from September to November 2019. The workshops were held from 10:00 to 14:00, and the participants spent their working hours participating.

The purpose of the workshop course was to increase the participant’s knowledge about the physical workload in the construction industry and to inspire, through these new insights, the finding of new solutions to reduce workload that they could use as work-environment professionals. The participants received an agenda by e-mail before each workshop. Futhermore, the agenda was presented before the start of each workshop and each participant was informed about the purpose of each workshop. At workshop 1, the researchers gave oral presentations about a participatory ergonomics intervention in the construction industry and the foreman’s role in rolling out new ideas, based on results from previous projects [13,14,25,26], and these were discussed with the participants. Furthermore, the researchers presented new research findings about physically demanding work situations, based on a more in-depth analysis of previous research from our group [13]. In brief, this analysis consisted of data from 80 full working-days from construction workers (bricklayers and concrete workers), measured with electromyography, IMU-sensors, and video. The data had previously been analyzed to find the peak working situations [13]. In the present study, we analyzed all working situations during the working days. Seven themes of working tasks were presented to participants at workshop 1 (Table 1).

These seven themes were selected to address key hazards for MSD, e.g., high repetiton, heaviness, and awkward postures. The participants considered and discussed the new knowledge, developed an individual action plan, and presented and evaluated it during the course. The work-environment professionals individually selected one of the seven themes from the new knowledge they wanted to work with in their action plan. They were asked to develop the plan between workshops 1 and 2. The participants could work on their action plans in small groups, which resulted in nine groups. At workshop 2, the environment professionals presented their action plans and received feedback from other participants and researchers.

Furthermore, researchers presented current knowledge, about barriers for the prevention of MSD and implementation of workplace safety knowledge in the construction industry, based on previous research [25,26]. After this, participants discussed how they could counteract these barriers before evaluating their action plan. At the third workshop, the participants presented and discussed their revised action plans. In addition, the workshop included presentations about “preventive strategies for implementing and maintaining initiatives for workers with hard physical work” and “successful work environment coordination”.

### 2.4. Evaluation of Workshop Course and Outcomes

#### 2.4.1. Questionnaires

The workshop course was evaluated at the end of the third workshop, using a short survey. The questionnaires were related to the participant’s satisfaction, content, teachers, and general workshop elements. As well as questions about the applicability of the workshop course, motivation to work with the new knowledge, if participants succeeded with implementing their action plans, and if the participants were confident in implementing their action plans in the future. The current research focused on the following questions: “To what extent have you succeeded in implementing your action plan at present?”, “To what extent do you have confidence that you will succeed in implementing your action plans in the future?” and “To what extent have you used the knowledge you gained from the course in your daily work, since your participation in the course?” The questions were answered with a response scale from one to five, where one was ‘to a very low degree’, two was ‘to a low degree’, three was ‘to a moderate degree’, four was ‘to a high degree’ and five was ‘to a very high degree’, as has been customary in the Danish Work Environment and Health study [1,27].

#### 2.4.2. Phone Interviews

Phone interviews were conducted approximately three months after the end of the workshop course and lasted approximately 30 min, and were based on questions about the participant’s specific action plan that had been prepared (and possibly implemented) during or after the workshop course. The interviews were carried out during regular working hours, as was most convenient for the participants, and were scheduled beforehand. The interviews were semi-structured and asked about relevance, motivations, opportunities, and barriers to using new knowledge from the workshop course. The interview guide was based on the questions presented in Table 2 and were about implementation, adaptation and effectiveness. The questions were answered with a response scale from one to five, where one was ‘to a very low degree’, two was ‘to a low degree’, three was ‘to a moderate degree’, four was ‘to a high degree’ and five was ‘to a very high degree.’

### 2.5. Analytical Framework

Empirical material from the phone interviews was initially categorized according to the three structured questions: (1) “to what degree did you succeed in implementing your action plan at the present time?”; (2) “to what degree do you think that you will manage to implement your action plan in the future?”; and (3) “to what degree have you employed the knowledge you obtained at the course in your daily work?”. All three questions were answered on a one-to-five point scale, where one was ’to a very low degree’ and five was ‘to a very high degree’. For these questions, we calculated averages, describing the generally perceived level of impact from the workshop program on each of the three measures.

As our interest is primarily in understanding how the participants used and incorporated knowledge in practice, the main part of the analysis consists of an agential realist-positioning analysis, which was used to analyze the interviews [26]. Positioning analysis in an agential realist framework works from the notion that phenomena are (re)configured (given meaning and characteristics, both in the understanding of humans and in their worldly functioning [28]), through both linguistic intra-actions between people communicating and in the intra-actions between language and materiality (bodies, organizations, work tasks, machines etc.). Positioning analysis was originally developed by Davies and Harré [29] and theorizes that people, in conversation and other language usage, ascribe characteristics to themselves reflexively and to others interactively. In this linguistic negotiation of characteristics, interlocutors may conform to positionings or they may reject them. Conforming to a positioning implies the interlocutor taking on the ascribed characteristic as part of their identity, at least in the sense that they may be recognized by that characteristic in the specific social context.

In practical terms, the analysis was conducted based on a coding of the empirical data aimed at examining the respondents’ positionings of the course, in the sense that they ascribed characteristics to the course and hence made claims about its effect in the world.

The analysis was subsequently divided into the three subthemes, which were the most commonly mentioned (re)configurations of the course: (1) the course as a means for networking and generating the ability to share knowledge; (2) improved knowledge about physical strain in construction work; and (3) applying OSH knowledge in practice. These themes are presented below, where their main tendencies are explained, and illustrative quotes/examples are provided. In the discussion, these themes are discussed with existing research.

## 3. Results

The participants in the study were OHS professionals, in this case defined as occupational health supervisors, work environment coordinators, work environment consultants, EHS Lead Managers, quality and work environment managers, project leaders, production managers, and foremen. Twenty OHS professionals started and seventeen completed the entire course and phone interviews. The flow of the study is presented in Figure 1.

For descriptive purposes, the action plans for the nine groups of participants are shown in Table 3.

Table 4 presents the answers from the evaluation questionnaires, conducted at the end of the third workshop, and connected with the phone interviews. In Table 5, the barriers to implementing the action plans are presented.

### 3.1. Analysis—Things That Work and Things That Don’t

The participant’s average score on the level of implementation of their chosen action plans was 3 immediately after the workshop program, and 2.5 at the follow-up. Scores of 3 or 2.5 out of 5 might suggest that actions had been initiated and some progress had been made—at least in some cases. Based on the qualitative evaluations, this did not, however, seem to be the case. In most cases, the action plans (briefly described in Table 3) had not been implemented. Some explained that the action plan had been cancelled, because a similar initiative was already in place. Some explained that they just had not acted on the action plan.
I: “So the action plan, how did it go with that?”
P: “Mmh, I mean. We didn’t really do as much as I wished. But sometimes, you just don’t get something… I mean all the things you want.”(interview person 45)

Here, the participant explains that they have just not gotten things done, without offering other explanation. The participant positions herself as rather powerless in offering the suggestion that sometimes you just cannott get what you want. This powerlessness, suggested as a reason for not implementing the participant’s action plan, is in agreement with previous research on OSH professionals. There, it has been described that OSH professionals often have a hard time gaining traction within their organizations, and that, often, they are encouraged to perform administrative, legitimizing, or socially-oriented tasks, rather than tasks that actually change the physical safety and health of workers [30]. Others described that their action plan was no longer relevant, as tasks had changed, and one argued that COVID-19 had come in the way of their initiative. Examples are shown in the following two statements:
“So that’s a 1 or 2 on that scale. And it’s been put to rest, because we had another project going at that point. And the poster we wanted to make was for workers, but we have another project targeting foremen and site managers, so those two would get in each other’s way.”(interview person 30)
I: “So, regarding the action plan, that’s a 2. Can you explain why?”
P: “I will say it was hard… I mean it requires that you have a task that fits with resolving this physical exertion. At the site we were at. There were loads of problems, but they concerned access roads and dust and cleaning. And then we were basically done.”

Fifteen (of seventeen) interviewed participants described similar situations for the lack of successful action-plan implementation. The remaining two participants did not specify exact action plans, even though course participants were asked to formulate concrete, practical plans and were urged to reflect upon particular actions to implement, barriers to account for, and who would be responsible for these actions.

On the other hand, participants positively (re)configured the course’s role in improving their knowledge about physical strain in construction work, and on applying this knowledge in practice, as OSH professionals on construction sites and in organizations. Also, they (re)configured the course as contributing to their networks and ability to share knowledge, as well as their knowledge and awareness of implementation practices. The following section explores the implications of this and exemplifies these qualities of the course.

#### 3.1.1. Improved Knowledge about Physical Strain in Construction Work

While few participants actually implemented action plans, almost all of them described having learned something new and useful from the course. As one participant described:
P: “I must say, that’s a five, because some of the information we had, they’re just stuck in my head. Particularly regarding the heavy lifting, that was an eye-opener. So that is something you focus on each time you have to perform heavy lifting, right. We were told, now I don’t remember exactly how many kilos it was, but the higher you lifted above the head—it rose several kilos at the time. And that’s thought provoking. Because normally you just do it, you always just did. So that’s a five.”
I: “So there were some things that surprised you?”
P: “yes, yes there were. I mean, you know that the burden is several times higher. But that it was so much, I did not have a clue. As I say, you always just did that if you had to. Then you just groaned a bit if it was really heavy. But that was really thought provoking.”(interview person 8)

In this, the participant positions himself as highly engaged in the course topic and knowledge, and as having obtained new knowledge that changed his orientation towards lifting. In this sense, the course is (re)configured as a phenomenon providing this knowledge. In this instance, the new knowledge concerning heavy lifting and that incrementally increasing weight, caused by working with your hands above the head, was specified as an eye-opening experience to the participant. This particular issue was a main point of new knowledge to many of the participants:
“What was an eye-opener to me was this thing about the kilos that grew immensely when working with the hands above the head. And that was actually the material that gave us the decisive knowledge to show a man that, when you lift the stone here, that’s not a problem. But when you do it a thousand times, then it starts. And I can understand that that is a form of strain. And there was that man from your observations—which was really great—and you started thinking; we may not be able to avoid lifting, but we may start telling people how they can avoid the repetitive strain and the outer positions.”(interview person 30)

As described, many participants (re)configured the course as a phenomenon providing knowledge on the exertion connected to working with the hands above the head. This participant made a point of acknowledging the repetitive character of the work, and connected the information provided at the course with new insight into the problem of lifting many times during a workday.

Several others pointed out that, even though they felt aware of the physical exertion in work before the course, a main point, for them, was obtaining the knowledge that scientific documentation now existed for this aspect of their work on improving OSH in their construction companies. As one of the participants pointed out, when asked how they think about participating in the course:
P: “I think it was really great, I hope there will be more of that sort. Because even though we know that for instance cutting bricks is physically straining and the workers experience that too, then research shows that it is. So in that way, you have your back covered a bit more in saying—we have to do something about this. Because that’s what science says. So I think that’s been super great!”(interview person 41)

Here, the participant positions herself as knowledgeable on physical exertion, but in need of support from the evidence to have their “back covered”. The course was positioned as providing this. Hence, providing new scientific knowledge for professionals on tangible subjects, such as the increasing exertion connected to working with the hands above the head, with repetitive work, and also providing scientific legitimacy to the OSH professionals’ claims, are (re)configured as two important course characteristics.

#### 3.1.2. Applying OSH Knowledge in Practice

Even though many participants did not manage to implement the action plans, 12 out of 17 described that they employed knowledge from the course in one form or another in their professional OSH practice. One example of this is described in the following:
I: “You chose to address the employees to create some awareness among them or what were you thinking?”
P: “yes, yes. Because when you walk around there as an OSH nerd. Then you talk about strain and repetitive movements and twisting and pulling and all that. And honestly you don’t think about all that when you’re out there and just have to bind iron. So that thing with gaining an understanding of what we really mean. And the one with the brick, that little exercise, where you have to feel what happens in the body when you twist, we used that, I used it for some workshops that I do. And that’s one of the exercises now. I bring a box of bricks’. Then they go to a group session and I’ve made four exercises where they lift closely to the body, do a twist and so on. That’s a source of some surprise. About how many muscles they can actually feel in those situations.”(interview person 30)

The participant here positions herself, and OSH professionals in general, as health and safety nerds, and workers as something else. The knowledge, in the shape of a course exercise, is (re)configured as a bridge to overcome the gap between the nerd and the other (the worker). In this case, the professional describes using a practical presentation method from the course for creating improved awareness about physical exertion among the workers in her work site. In similar ways, other participants described employing project information as tools for bridging the gap between OSH professionals and workers; tools for addressing physical exertion in work to the workers in an attempt to emphasize this this problematic issue.

Others placed a great weight on the possibility of employing knowledge from the course in their daily practice:
“The things I heard at the course I’ve brought along when I’ve been out on the sites and seen them do different physical things they have to do. And I say’ you’re doing that wrong, listen up, you’re exerting your-self this and that much’. And that’s an eye-opener to people, when you say that. So I would like a little folder telling them where the exertion is hardest and where I can see what causes the deterioration. […]”
I: “What did they say [to that]?”
P: “They actually get very surprised about how much they’re physically strained, because they don’t feel like it’s that hard. So they get surprised.”(interview person 47)

This participant positions herself as a corrector of worker’s practice of exerting their bodies through work. In this function, knowledge of the appropriate way to work so as to avoid over-exertion is described as an important tool. The technical knowledge communicated during the course, on the extent of worker exertion during work, is positioned as important for OSH. As may be seen from the participant’s description, knowledge gained from the course has helped her to be more specific in describing the relationship between present physical practice and the potential long-term effects on her manual workers’ health. In this sense, the course is (re)configured as a contributor for amplifying the impact of the OSH professionals’ claims, since these were perceived as gaining more support.

Others took the method employed at the meeting as a means for developing their own practice. One participant elaborates on how they altered the knowledge from the courses to be used in the particular workplace:
I: “What was it that worked for you?”
P: “It was the way in which the course was made, so that we actively had to work with something that could be used in our company, something that was very hands on. You could quickly see a method for—even though it did not have to be a particular method—but you could quickly see, hmm I could do this in my company as well. So I mean, it was equal parts information, and then laying up to—what do we do ourselves. I think that was really, really great.”(interview person 32)

Here, the OSH professional positions herself as an innovator of methods for improving OSH, and, in this role, the course is positioned as providing tools for this innovativeness. Here the course is (re)configured as a generalizable method for developing other OSH initiatives, not only those applied at the course. The OSH professional addresses the hands-on orientation of the course. The fact that participants worked with an actual issue, relevant to their own practice, is thus drawn forth as a positive source of learning. This participant (re)configured the course as providing a new methodology for working with OSH issues in her organization. The point that participants had to be highly active in providing something that was usable was described as inspiring.

#### 3.1.3. The Course as a Means for Networking and the Ability to Share Knowledge

The course was also widely (re)configured as contributing to improving participants’ professional networks, and their ability to share knowledge across different organizations.
P: “You also obtain some knowledge from talking to others and by hearing how the different companies handle the issues, and what challenges they have. And you could say that sharing is always good, and you always gain something—no matter what. So that part was really good.”
I: “so what did it mean to you that this was a course with other OSH professionals?”
P: “I mean, that was super great. I feel that most of the companies that were there we are both colleagues and competitors. In the way that, we compete for jobs but in many situations we work together out on sites, and we employ each other as subcontractors in other situations. And I mean, on OSH that’s not a competitive parameter, and the more aligned we are on OSH the better. Because that makes it easier for the workers to navigate as well, because they change working places too. And I don’t think that workers health should be a competitive factor. So in that sense, sharing is really good.”(interview person 9)

The participant in this example positions the OSH professional as being in need of discussing matters with other professionals outside their organization. We know, from other research on OSH professionals, that this is imperative to performance as an OSH professional, because OSH professionals often experience a lonely role in their companies [31]. The participant further positions the course as an entity providing the opportunity for sharing knowledge about workers’ health. Quite a few participants described going to other participants with open questions or potential solutions, and being receptive to their professional commentaries, as a great way of learning something new and useful for their own practice. Some even cooperated on specific practical problems, helping each other out on their construction sites.

To others, the chance of being recognized and not feeling alone with the challenges of improving OSH in construction work was important:
“[The course] worked really well. I mean. You could feel that we share almost the same challenges. I mean, we’re all thinking; oh ’’m so frustrated with this not going the way I want it to. And I could hear from the others that they share the same experience. So in that sense, with an OSH lens, the grass was not much greener in the neighbours’ yard. And that part worked fine too, because then you sit with a group that share the same challenges, and who wants to change the situation, so that’s really good actually.”(interview person 31)

Here, the participant positions the OSH professional as a person who, at times, experiences a hard time getting successful OHS outcomes. In this, the course, as a place to meet and discuss with other professionals, is (re)configured as a platform for gaining consolidation. This was mentioned several times as an important gain from participating:
P: “You meet others who have challenges and you provide some good ideas.”
I: “So you inspire each other?”
P: “You do, and you get consolation, that this can be really hard.”(interview person 34)

This participant repeats the above positioning of both the OSH professional in construction, and of the course as a platform, for gaining consolidation. Several of the participants pointed out the importance of meeting others who have the same challenges, and maybe even in providing some new good ideas. One participant even expounded that it can be a consolation, knowing other similar companies and OHS professionals have the same issues. As such, the course was (re)configured as contributing to improved knowledge-sharing across organizations, and also, as improving networks among OSH-professionals who, through participation, gain access to the knowledge of professional colleagues’ expertise in OSH solutions for resolving problematic issues, and also as a network that may relieve some of the stress of being an OSH professional in an industry where it is very hard to obtain tangible progress and results.

#### 3.1.4. Improving Knowledge and Awareness of Implementation Practices

Some of the issues that participants also noted were a focus on implementing OSH measures and what type of initiatives may help to achieve that:
“Well as OSH professionals we’ve been really good at making a poster or a flyer or whatever. And then we believed it would change people’s behaviour, but you don’t. And all research shows that. So this thing where you affect people in different places and at different levels—that’s what changes something. And then we have to pay attention to—I mean the comments you have been out getting in your observations. I mean how little the worker actually dares to suggest in terms of alternative ideas. That was very interesting.”
I: “You mean, so they challenge the foreman or the colleagues?”
P: “I mean, your research showed that if the foreman has an old-school conservative attitude, then the workers interest in providing suggestions, they die. And that’s why we need a focus on, how do we manage our workers. And the practical experience, we can’t take that away from the foreman, but we can talk about how we lead people. And that leadership has perhaps primarily been on the office side of things not on the foremen.”(interview person 30)

This participant positions OSH professionals as being capable of employing particular methods for behavioural change. The participant further positions the course knowledge obtained as a way of (re)configuring the OSH professionals’ ability to affect change, on the basis of new knowledge. In this case, the participant referred to knowledge presented during the course, on foremen’s identities and their resistance to being challenged on their managerial decisions, regarding OSH. This is a (re)configuration of the course as an entity that allows OSH professionals to reflect on, and improve their ability to “say the right things to the right people”, which has been described as important for OSH professionals in their aims to influence health and safety [32]. Also, some participants felt closer to making an impact on OSH legislation and the creation of knowledge.
P: “I think it could be a lot of fun to do some research—and legislation for that matter. And here you see that there’s actually a short way from the construction site to that legislation. And then all of a sudden, it makes sense.”
I: “Sure, how do you see yourself in that as an OSH professional?”
P: “I see myself as the best possible way to transfer that out there. I believe I have good prerequisites for that, because I’ve been out on the scaffold laying bricks myself. I actually know what I’m talking about. I know how it feels. And all of a sudden, I’ve got the numbers and measurements on what we’re doing to ourselves. [our company] we don’t have that much bricklaying, but it’s not so hard to convert that knowledge to other tasks.”(interview person 34)

As described here, the participant positions himself as highly engaged in the prospects of improving OSH and as an expert on bricklaying practice. Further, he positions the course as having the potential to create a link between knowledge, research, and even legislative practices. In a similar way, other participants (re)configured the course as a way of closing the gap between workers and lived working life on construction sites and the scientific creation of knowledge; and, through that, affecting the political decision processes. Additionally, the participant again learns something that may be important at a broader level than the specific matter at hand.

Another participant mentioned the course’s role in educating champions or knowledge-brokers to spread knowledge into construction organizations.
I: “Yes what was your impression of us presenting that knowledge to some OSH professionals?”
P: “I think that was really good. I mean changes don’t happen by themselves. Someone—you could call them ambassadors, or whatever… but I mean, someone has to implement this knowledge out in the companies. So if you can get someone to interest themselves primarily with these things in practice out in the companies. Someone that you educate particularly so they can contribute to change reality out in the companies. It won’t happen in one day. But that is one of the ways.”
I: “Did you use some of that knowledge in your work?”
P: “Yeah sure. It was confirmed for me that we have to do something about tying rebar. So it’s not something I talk about daily. But almost daily—when we discuss the OSH issues connected to concrete work.”(interview person 39)

This shows the participant positions the course as an enabler of the creation of OSH ambassadors, and their view that these are needed for change. As such, the course becomes an intermediary OSH measure for implementing other measures at higher oranizational levels, e.g., ambassadors.

Another participant indirectly addressed the course’s potential for contributing to the implementation of OSH measures, as she mentioned her own approach to the course in her organization:
“Myself and these three guys. We met and drew some stuff and so on. And discussed what we wanted to do, and how we wanted to do it. Maybe it should have been more like, but the situation made it a bit hard, but perhaps it should have been like; by participating in these workshops, you have to put aside half a day to fulfill or handle these issues. We wanted to make some prototypes of some lifting tables and stuff, right. And that wouldn’t have been unrealistic at all, to do that in a couple of days. So we could have planned that from the beginning. So the foreman of these guys, so he could plan with them participating a few days between the workshops too. Then I think we would have achieved it.”(interview person 45)

Several of the participants reflected that only a few had taken the opportunity of making the course into an experience for both workers e.g., OSH representatives, and OSH professionals. Two companies took the approach of engaging both, and their participants described that, even though workers have a hard time staying awake for 4 h workshops with lots of talk and sitting still, they felt that the engagement of both hands-on knowledge of work, OSH knowledge, and scientific knowledge were a potent combination for imagining new solutions.

## 4. Discussion

The present study examined how participants used and incorporated research-based knowledge from a three-day workshop course into practice. Questionnaires and phone interviews were used to collect data, which was analyzed using positioning analysis.

Positioning analysis in an agential realist framework contends that not only the characteristics of people and their social identities are negotiated in positioning situations. Also, the characteristics of other phenomena, such as organizational characteristics, the physical character of work, and structural barriers for improving safety and health at work, are negotiated through positioning. Thus, when an OSH professional, in an interview says, “for instance, cutting bricks is physically straining and the workers experience that too, then research shows that it is. So, in that way, you have your back covered a bit more” she ascribes certain characteristics to “research” as being legitimizing of practice, which may be negotiated by interlocutors, drawing on their knowledge of or opinions on the subject, or stand unchallenged and thereby remain a particular (re)configuration of the workings of the world [26]. This should not be mistaken for a social-constructionist perspective, claiming that all knowledge claims are equally valid. Rather, both method and theory may be drawn into the discussion of the characteristics of people and phenomena; many knowledge claims may be contested or rejected by better, more thorough, more precise, or complex measuring methods. If they are not, however, this approach implies that knowledge is also always a locally negotiated matter, with implications for the orientations and actions of people. Even though better knowledge, methods or technology may exist somewhere in the world, people act on and negotiate their knowledge and conduct based on their own perceptions and available technology. In relation to our subject matter, this means that the ways in which OSH professionals position the workshop course, their organization, and the construction industry as a whole, will tell us a lot about how a course of this kind may affect OSH professionals’ abilities to translate knowledge into changes in OSH practice, and what to be particularly aware of when seeking to do this.

The course employed in this study consisted of evidence from scientific research presented by researchers, with course participants encouraged to select relevant knowledge and develop an action plan for implementation in their respective workplaces. The course was delivered at no cost to the study participants, and was well attended, despite the time demands (time to complete the course and how busy they were). Overall, participants reacted positively to the workshop course. The positives revolved around having improved knowledge about heavy work hazards and applying it in their workplaces. They also noted the value of networking with other OSH professionals. The participants (re)configured the course as improving their knowledge about physical strain in construction work, focusing on applying this knowledge in practice, as OSH professionals, on construction sites and in organizations. Furthermore, they (re)configured the course as contributing to their networks and abilities to share knowledge, and to their knowledge and awareness of implementation practices. Maintaining an excellent professional network and developing an understanding of how to change organizational behaviors have been shown to be essential for OSH professionals who work to improve OSH (Provan et al., 2017). Furthermore, participants considered the knowledge presented as useful in their day-to-day work. They also described how they used the knowledge gained from the workshops at their workplaces. This use of evidence-based knowledge is positive and may reduce hazards in the workplace. In this sense, these effects may be interpreted as positive effects gained from the training course and are assumed to be intermediate factors for action on the work sites. This is further underlined by Morris et al. (2011), in a literature review describing and quantifying time lags in the health research translation process, which showed there can be a lag in the research translation process of up to 17 years [33]. Therefore, there may also be a considerable time lag in getting new OSH knowledge and practice implementation to working sites. The intermediate factors, shown in the present study, could be interpreted as a very positive finding.

In a rapid review, Van Eerd (2019) found that the literature suggests three principles to include in developing a knowledge transfer and exchange approach for health and safety: (1) include workers as a target audience; (2) involve a researcher as a disseminator; and (3) include in-person meetings and printed materials in a multi-activity knowledge transfer and exchange plan [34]. Moreover, Meissner et al. 2020 describe the inclusion of stakeholders as “[d]one correctly with sensitivity, inclusion, and respect, it can significantly facilitate improvements in research prioritization, communication, design, recruitment strategies, and ultimately provide results useful to improve population and individual health” [35]. The present study followed these principles to a great extent. Even though the participants were mostly OSH professionals, they all had direct contact with workers on construction sites in their daily work. They served as the appropriate target audience for this workshop program. Researchers disseminated the new knowledge to the participants. The development of their action plans was based on multi-activities, since there was both visual and verbal knowledge transfer from video clips, presentations, and discussions, respectively. Therefore, the workshop course could have sowed the seeds for future changes in the construction industry.

Even though most action plans were not deployed in practice at the time of evaluation, the participants had great faith that they would implement their action plans in the future (Table 4). There may be several explanations for the participants not implementing the action plans. While the action plans were part of the workshop course, it was not certain that participants could implement the plans within three months. However, we encouraged them in their attempts to implement their action plans.

The construction sector is complex and dynamic, both in reporting relationships [19] and in organizational decision making [36]. The OSH professionals who participated in the present study may not be able to unilaterally ensure that action occurs at the workplace. They are agents of change, but not decision-makers. Hence, they have competencies in OSH but may be unable to require or impose changes in practice. However, studies have shown there are pathways in the construction sector where knowledge can be transferred. Carlan et al. (2012) suggest flexible and multi-directional lines of communication must be used to transfer knowledge in the industry [19], and Yanar et al. 2019 suggest that providing OHS leaders with practical information and tools can help in decision making and potentially improve OHS performance [36]. Furthermore, Sinelnikov et al., in a 2020 review, found consistent evidence for the effectiveness of supervisory training interventions across several outcome measures [37]. The present study engaged OSH professionals, which included supervisors, foreman, consultants, and the Danish work environment authorities, and this group of participants seems to make decisions that could affect the physical workload at the worksites in a positive way.

The present study’s participants reported a lack of time as the primary barrier for not implementing their action plan (Table 5). This finding is in accordance with previous research [38,39,40]. A qualitative study, incorporating survey, interview, and focus group data collection, studied the experiences and perspectives of OSH knowledge-users about research use, and found that implemented new findings from the research were added to their regular workload [41]. Our participants were delighted with the training course. The possibility of speaking about and discussing the challenges of physically demanding work tasks is mentioned as being encouraging in the interviews. The possibility of communicating within peer networks and sharing research with colleagues, even if they were competitors, was also found in the literature as a positive finding [21,41]. Therefore, providing a training course that included group work and participatory activities might be a good approach for transferring new knowledge to OSH professionals; participants could get easy access to knowledge and transfer their research to business.

The training course in the present study was provided free of charge, and with a limit of 20 participants. That it was free of charge could have influenced the motivation for participation. The course was, quickly, fully subscribed to, and we established a waiting list. Even though it was very positive that the training course gained a lot of notice, as it was fully attended, and we did not see a considerable drop out during the training course; (Figure 1) this could have been a bias that it was free of charge. The motivation from the participants to implement their action plans could have been higher if they had paid for a training course.

Construction companies’ core task is profit, which may be prioritized to a higher extent than a given project; this may have influenced the implementation of participants’ action plans [26]. Furthermore, we, as course leaders, could have been more explicit in our expectations to the participants before the start of the training course. While it is always possible to improve on workshops, the feedback from participants suggests that the workshop was valuable. As the participants were OSH professionals, they were familiar with the topics, on a certain level, from previous training.

We only included large-scale construction companies, and different results might have occurred if small-scale construction companies were included. Thus, the generalizability of the results is limited to large construction companies. Furthermore, construction is an industry that is very dynamic and features a lot of variation, therefore, the generalizability of the results to other businesses might be challenging. However, the OSH professionals were indeed satisfied with the workshop course, so it might be a good approach for introducing new knowledge in other businesses. This study only included data and OSH professionals from the construction industry. Therefore, we have not examined whether or not the results would have been the same for other occupational groups, e.g., white collar workers.

## 5. Conclusions

The present study examined how participants used and incorporated research-based knowledge from a three-day training course in practice. Data were obtained from questionnaires and phone interviews. The participants were satisfied with the workshop course and the possibilities to increase their knowledge through new input from researchers and colleagues from other construction businesses. In particular, the qualitative study revealed that participants felt they had improved their knowledge about physical strain in construction work and how to apply OSH knowledge in practice, and had improved knowledge and awareness of implementation practices. Also, they reported that the course was a great means for networking and sharing knowledge. The participants reported that they believed that they would succeed in implementing their action plans in the future. Based on these findings, it is our conclusion that participatory workshops are a good approach for transferring research evidence such that end users can apply it, since participants can learn from each other and discuss challenging problematics with colleagues from other companies. Whether and how they use it likely requires more research. The present results provide us with a better understanding of how OSH professionals adopt new knowledge about MSD and physical workload, and how they use it in their daily work. This can be valuable information to companies in the construction industry aiming to secure the best possible knowledge for their OSH professionals.

## Figures and Tables

**Figure 1 ijerph-18-08498-f001:**
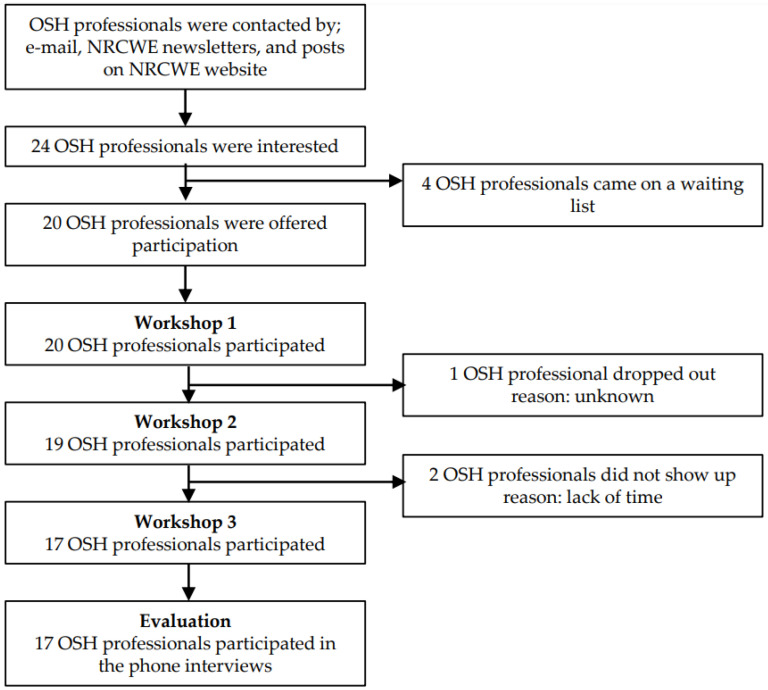
Flow of the participants in the study.

**Table 1 ijerph-18-08498-t001:** Concepts from workshops with areas of concern. All themes were presented on workshop 1.

Theme	Work Task
Manual material handling	Lift of material
	Lift of rebar
	Lift of bricks with brick tong
	Carrying of material
	Carrying of rebar
Reinforcement	Iron binding
	Manual handling of rebar
	Separates rebar from each other
Concrete forms	Push and pull of concrete forms
	Lift and carry concrete forms
Wheel barrow	Push of wheel barrow
Push of wheel barrow	Hip—shoulder height
	Shoulder height
	Above shoulder height
Bricklayer’s assistant	Focus on bricklayer’s assistants diverse work tasks
Focus on bricklayer’s assistants diverse work tasks	Casting with hose from casting bucket
	Vibration of concrete

**Table 2 ijerph-18-08498-t002:** Questions from the phone interviews. * The questions were the same as in the questionnaires.

Theme	Question
Implementation *	To what extent have you succeeded in implementing your action plan at present? To what extent do you have confidence that you will succeed in implementing your action plan in the future?
Adaptation	To what extent have you used the knowledge you gained from the course in your daily work, since your participation in the course? What knowledge have you used? a How did you apply the knowledge you gained from the course? b Why did you apply that particular knowledge?
Effectiveness	(How) have you passed on that knowledge from the workshop course to colleagues/management/industry communities? (How) has the knowledge you gained during the workshop course affected the work environment/the extent of the physically stressful working positions at your workplace?

**Table 3 ijerph-18-08498-t003:** Action plans for the nine groups of participants.

	Workshop Course		Evaluation	
Group	Issues	Action Plan	Implemented Action	Barriers
1	Work tasks that put more strain on the body than expected.	Poster for the employee with focus on lifting close to the body.	They are still planning to do the poster, but because company is running another project, the poster is on hold.	Organizational changes.
2	Handling of rebar.	Reduction of manual handling and heavy lifting when laying rebar, as well as reducing awkward working postures.	Plan to implement ergonomic education on the construction sites.	Time, start on new construction sites.
3	Work with drilling horizontal holes in foundations.	Reduce physical strain during drilling.	Contacted foreman from the bridge department, contacted a consultant, about assistive devices, bought vibration-damping gloves, drill carriage could be suitable for interior work, contact and visit by a consultant and subsequent evaluation on site, the construction workers were very satisfied with the consultant’s visit and thereby gained an explanation and understanding of how much they have to drill to stay within the limit value.	Lack of technical solution to the problem.
4	Material transport.	Improve access roads for the benefit of all groups of construction workers.	Increased attention to cleaning up.	Organizational changes—working on different construction sites.
5	Heavy lifting and awkward working postures.	Develop/introduce an assistive device with adjustable working height.	Increased focus on heavy lifting and awkward working postures.	Economy and lack of time.
6	Casting of concrete slabs e.g., bridges and working with lack of space.	Prevent pulling and manual handling of vibrator hoses and improve access roads.	Working on instruction videos to increase the reach within the company.	Lack of time.
7	Paving work, pavement is crushed with hammer and collected manually.	Reduce bad working posture with twists and bends in the back, hips and knees.	Tested various technical devices.	Lack of technical solution to the problem.
8	Reinforcement work.	Short course in lifting technique. Toolbox talks with focus on increasing the use of assitive devices. Instruction videos, with a fokus on lifting tecknique, exercises for warming up, and for exercises during the working day. Posters with data on risk factors.	Toolbox talks, with a focus on increasing the use of assistive devices. Still planning to do the videos, but is delayed due to the pandemic situation.	Organizational changes.
9	Reduce the physical load when cutting bricks.	Develop/introduce an electric stone cutter.	Developed a prototype of an electric stone cutter, and plans have been made to test it.	Lack of time.

**Table 4 ijerph-18-08498-t004:** Evaluation from questionnaire.

	After Workshop		Follow Up	
Question	N	Average Score	N	Average Score
To what extent have you succeeded in implementing your action plan at present?	14	3.00	16	2.50
To what extent do you have confidence that you will succeed in implementing your action plan in the future?	15	3.80	14	4.14
To what extent have you used the knowledge you gained from the course in your daily work, since your participation in the course?	16	4.25	15	3.57

**Table 5 ijerph-18-08498-t005:** Barriers for implementing the action plans.

	Have Any of the Following Factors Been a Barrier to Your Work on the Action Plan (You May Tick Off More Than One)?	Do You Think any of the Following Factors Will Be a Barrier to Working on Your Action Plan (You May Tick off More Than One)?
Lack of time	9	5
Lack of financial resources	1	1
Organizational changes	2	2
Resistance from employees	0	1
Resistance from the leaders	2	1
No barriers	3	6
No Answer	1	1

## Data Availability

Data can be available upon reasonable request.

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
