# Peer review of "Engaging Occupational Safety and Health Professionals in Bridging Research and Practice: Evaluation of a Participatory Workshop Program in the Danish Construction Industry"

_ijerph, 2021, doi:10.3390/ijerph18168498_

Round 1

Reviewer 1 Report

The authors were very responsive to this reviewer.  The manuscript is much improved for readability, has been condensed somewhat and organization is improved.  Overall the paper is very much improved.

Author Response

Thank you for your feedback and your time used for reviewing the manuscript

Reviewer 2 Report

Title

The title is unclear and does not reflect the research. The title is very interesting for a television documentary. For this research, it is not clear what the title means. It is important that the title has at least three keywords.

Abstract

In the summary, it is not necessary to identify the different sections. It is recommended to remove the words: background, aim, methods, results and conclusion.

Remember that the maximum number of words that the journal allows in the abstract is 200.

Keywords

Regarding keywords, it is very surprising that the term R2P is raised. R2P means:

"The Responsibility to Protect (R2P or RtoP) is a global political commitment which was endorsed by all member states of the United Nations at the 2005 World Summit in order to address its four key concerns to prevent genocide, war crimes, ethnic cleansing and crimes against humanity."

"About the Responsibility to Protect". www.globalr2p.org.

R2P and Responsibility to Protect are not mentioned in the text of the article. Please remove the term "R2P" from the keywords.

Modify the keyword "knowledge transfer and exchange" with the keyword: "participatory workshop"

Delete the keyword "Muscular skeletal disorders" as it is only mentioned in the first paragraph of the article introduction. This keyword does not identify your research. If you consider keeping this keyword, it would be interesting for you to draw conclusions in its corresponding section regarding the MSD.

Delete the keyword "blue-collar" as it is not mentioned in the article and is not appropriate. All workers have to comply with and enforce the regulations on the prevention of occupational hazards; blue-collar and white-collar workers.

Please, identify the article with keywords appropriate to your research. I suggest the following keywords: "training and information", "occupational risk prevention", "construction workers"; or similar.

  1. Introduction

  1. Materials and Methods 80

2.1 Participants 81

It is important that you identify the nationality of the workers.

2.2 Ethics 92

No comments.

2.3 Workshop course 96

It is important that Table 1 identifies the workshops and their corresponding themes. You can add a column to the left of the table.

It is also important that you explain the time of each session.

Is it sure that all workers have understood the reason for the sessions and their meaning?

2.4 Evaluation of workshop course and outcomes 133

2.4.1 Questionnaires 134

Please justify why the scale of values is five possible answers. Also identify the five possible answers.

It is not correct to propose a scale of values from 1 to 5. I understand that the values were 1-2-3-4-5. They are odd and I recommend that for future research, raise a scale of even values: 1-2, 1-2-3-4; 1-2-3-4-5-6; 1-2-3-4-5-6-7-8; etc. In this way there is no middle ground. normally, workers tend to go to the core values due to ignorance.

2.4.2 Phone interviews 147

Did the phone calls take place throughout the all day or were they concentrated at the same hours? Were the questions asked in the worker's home or in the work environment? The worker was asked about his individual action, but was he asked about his collective action or participation in prevention? Did the workers have previous prevention training? What was the level of previous training?

The circumstances above in which the phone calls were made should be clarified.

2.5 Analytical framework 168

Es importante que liste las cinco categorías de respuestas.

In Table 3, the right column identifies some barriers in the workshops. Why was there a lack of time? It is important that you add a column that reports on the full time spent in the workshop. What schedule was followed to carry out the workshops? Was it outside of work hours or during work hours? This could explain the lack of time. Could this lack of time be due to poor workshop planning?

all the above issues have to be clear.

  1. Results

It is important to explain what type of companies the workers who participated in the workshops belong to. Where the construction sites are located (geographically). It is important that you provide photographic information of the works in which the workers work when the survey was carried out.

Table 3 is not correctly framed on the page and it is not known if any columns are missing.

3.1. Analysis - things that work and things that don't 249

Lines 256-257: "Some explained that the action plan had been canceled, because a similar initiative was already in place. Some explained that they just had not acted on the action plan." Please give a further explanation of this comment. Was the company unaware that this investigation was being done? Did the company managers sign and accept the informed consent?

Lines 259-262: "I: 'So the action plan, how did it go with that?' 259

Q: 'Mmh, I mean. We didn't really do as much as I wished. But sometimes, you just don't get something… I mean all the things you want. '"You cannot allow the reader to draw a subjective and random conclusion from this comment between the interviewer and the worker. Please explain what it means for research this comment contributed by the worker.

3.1.1 Improved knowledge about physical strain in construction work 294

3.1.2 Applying OSH knowledge in practice 349

3.1.3 The course as a means for network and ability to share knowledge 418

3.1.4 Improving knowledge and awareness of implementation practices 476

What do all the comments raised in all section 3.1 contribute to the investigation?

Please explain the term "(re) configured"; it is used very often in the text.

Please explain how the workshops were carried out, how the workshops were structured, the agenda, questionnaires, tasks, questions. Did the workers have a paper agenda or was it all a powerpoint presentation? Where were the workshops held, in a company room or in a place set up for such an event?

The workshops have been raised for higher category workers (white-collar). I explained why the workshops for lower-ranking workers (blue-collar) have not been held.

  1. Discussion
  2. Conclusion

Please expand the conclusions. They are very poor for the research that has been done. Surveys with qualitative values and quantitative values have been obtained, so the conslusions can be extended further.

Indicate what implications this type of workshops can have for the improvement of health and safety conditions in construction companies.

Please avoid paragraphs with only one sentence.

References

There are 41 references in total and 10 current references (between the years 2018, 2019, 2020 and 2021). This equates to 24% of current referrals. They are, relatively, few updated references. It is convenient that you update the bibliography until you get 30% of current references.

The dates of each reference must be in bold.

Author Response

Thank you for your feedback and your time used for reviewing the manuscript. We really appreciate it, and thinks the manuscript is approved based on your feedback. Our replies will appear in the attached file.

Best regards

Mikkel

Round 2

Reviewer 2 Report

Title

The title is now much clearer and more accurately defines the reason for the investigation. But it is important that the country is added.

Abstract

Thanks for the modifications. The Abstract is correct and clear.

Remember that the maximum number of words that the journal allows in the abstract is 200.

Keywords

Thanks for the modifications. The keywords are adequate and correct.

  1. Introduction
  2. Materials and Methods 80

2.1 Participants 81

It is important that you identify the nationality of the workers. It is very rare that they are all Danish.

2.2 Ethics 92

No comments.

2.3 Workshop course 96

It is important that Table 1 identifies the workshops and their corresponding themes. You can add a column to the left of the table.

It is also important that you explain the time of each session.

Is it sure that all workers have understood the reason for the sessions and their meaning?

2.4 Evaluation of workshop course and outcomes 133

2.4.1 Questionnaires 134

Please justify why the scale of values is five possible answers. Also identify the five possible answers.

It is not correct to propose a scale of values from 1 to 5. I understand that the values were 1-2-3-4-5. They are odd and I recommend that for future research, raise a scale of even values: 1-2, 1-2-3-4; 1-2-3-4-5-6; 1-2-3-4-5-6-7-8; etc. In this way there is no middle ground. normally, workers tend to go to the core values due to ignorance.

2.4.2 Phone interviews 147

Did the phone calls take place throughout the all day or were they concentrated at the same hours? Were the questions asked in the worker's home or in the work environment? The worker was asked about his individual action, but was he asked about his collective action or participation in prevention? Did the workers have previous prevention training? What was the level of previous training?

The circumstances above in which the phone calls were made should be clarified.

2.5 Analytical framework 168

It is important that you list the five categories of responses.

In Table 3, the right column identifies some barriers in the workshops. Why was there a lack of time? It is important that you add a column that reports on the full time spent in the workshop. What schedule was followed to carry out the workshops? Was it outside of work hours or during work hours? This could explain the lack of time. Could this lack of time be due to poor workshop planning?

all the above issues have to be clear.

  1. Results

It is important to explain what type of companies the workers who participated in the workshops belong to. Where the construction sites are located (geographically). It is important that you provide photographic information of the works in which the workers work when the survey was carried out.

Table 3 is not correctly framed on the page and it is not known if any columns are missing.

3.1. Analysis - things that work and things that don't 249

Lines 256-257: "Some explained that the action plan had been canceled, because a similar initiative was already in place. Some explained that they just had not acted on the action plan." Please give a further explanation of this comment. Was the company unaware that this investigation was being done? Did the company managers sign and accept the informed consent?

Lines 259-262: "I: 'So the action plan, how did it go with that?' 259

Q: 'Mmh, I mean. We didn't really do as much as I wished. But sometimes, you just don't get something… I mean all the things you want. '"You cannot allow the reader to draw a subjective and random conclusion from this comment between the interviewer and the worker. Please explain what it means for research this comment contributed by the worker.

3.1.1 Improved knowledge about physical strain in construction work 294

3.1.2 Applying OSH knowledge in practice 349

3.1.3 The course as a means for network and ability to share knowledge 418

3.1.4 Improving knowledge and awareness of implementation practices 476

What do all the comments raised in all section 3.1 contribute to the investigation?

Please explain the term "(re) configured"; it is used very often in the text.

Please explain how the workshops were carried out, how the workshops were structured, the agenda, questionnaires, tasks, questions. Did the workers have a paper agenda or was it all a PowerPoint presentation? Where were the workshops held, in a company room or in a place set up for such an event?

The workshops have been designed for higher category workers (white collar). Explain why the workshops for lower-ranking workers (blue collar) have not been held. It is not understood why the keywords use "blue collar" when the research has been carried out on the highest professional category (white collar). It is recommended to remove the term "blue collar workers" from the keywords or change the name of the keyword to "white collar workers". If not, please explain what were the professional categories of the workers who participated in this research.

  1. Discussion
  2. Conclusion

Please expand the conclusions. They are very poor for the research that has been done. Surveys with qualitative values and quantitative values have been obtained, so the conclusions can be extended further.

Indicate what implications this type of workshops can have for the improvement of health and safety conditions in construction companies based on the results obtained.

References

Thanks for the modifications. The References section is correct and clear.

Author Response

Thank you again for taking your time for reviwing the manuscript. We really appreciate your work with the review. Some of the questions were the same as in the the previous revision, so in case you did not recieved our answers I have listed our answers below.

Best regards

Mikkel

Title

The title is now much clearer and more accurately defines the reason for the investigation. But it is important that the country is added.

Our answer: We have added contry in the title: ”Engaging occupational safety and health professionals in bridging research and practice: Evaluation of a participatory workshop program in the Danish construction industry”

Abstract

Thanks for the modifications. The Abstract is correct and clear.

Remember that the maximum number of words that the journal allows in the abstract is 200.

Our answer: Thank you. We have double-checked and the abstract is 199 words

Keywords

Thanks for the modifications. The keywords are adequate and correct.

Our answer: Thank you

  1. Introduction
  2. Materials and Methods 80

2.1 Participants 81

It is important that you identify the nationality of the workers. It is very rare that they are all Danish.

Our answer: They were all Danish. We have added this information in line 89

2.2 Ethics 92

No comments.

2.3 Workshop course 96

It is important that Table 1 identifies the workshops and their corresponding themes. You can add a column to the left of the table.

Our answer: All themes were presented on workshop 1. We have specified this in line 113 and 116.

It is also important that you explain the time of each session.

Our answer:  We have added this information in line 97.

Is it sure that all workers have understood the reason for the sessions and their meaning?

Our answer: Good point. We did explain the purpose of the workshop course and each workshop. But it could have been more explicit. We are mentioning this in the discussion section in line 685: Furthermore, we, as course leaders, could have been more explicit in our expectations to the participants before the start of the training course

2.4 Evaluation of workshop course and outcomes 133

2.4.1 Questionnaires 134

Please justify why the scale of values is five possible answers. Also identify the five possible answers.

Our answer:  We have justified the use of the scales in line 159 and information about the possible answers has been added in line 158.

It is not correct to propose a scale of values from 1 to 5. I understand that the values were 1-2-3-4-5. They are odd and I recommend that for future research, raise a scale of even values: 1-2, 1-2-3-4; 1-2-3-4-5-6; 1-2-3-4-5-6-7-8; etc. In this way there is no middle ground. normally, workers tend to go to the core values due to ignorance.

Our answer:  Thanks for the comment, we will have to take that into account in the future.

2.4.2 Phone interviews 147

Did the phone calls take place throughout the all day or were they concentrated at the same hours? Were the questions asked in the worker's home or in the work environment? The worker was asked about his individual action, but was he asked about his collective action or participation in prevention? Did the workers have previous prevention training? What was the level of previous training?

The circumstances above in which the phone calls were made should be clarified.

Our answer:  Information about phone interviews has been clarified in section 2.4.2.

As the participants were OSH professionals they were familiar with the topics on a certain lewel from previous training. This has been clarified in the discussion section in line 698.

2.5 Analytical framework 168

It is important that you list the five categories of responses.

Our answer:  We have added this information in line 168

In Table 3, the right column identifies some barriers in the workshops. Why was there a lack of time? It is important that you add a column that reports on the full time spent in the workshop. What schedule was followed to carry out the workshops? Was it outside of work hours or during work hours? This could explain the lack of time. Could this lack of time be due to poor workshop planning?

all the above issues have to be clear.

Our answer: This information has been added in section 2.3. We will ask the editor for the possibly for including the agendas in an appendix.

We see your point about poor workshop planning, however the participants were very satisfied with the programme. As stated previously we could have been more explicit in our expectations to the participants. We have added the following on that note in the discussion, line 707: “While it is always possible to improve on workshops, the feedback from participants suggests that the workshop was valuable”.

  1. Results

It is important to explain what type of companies the workers who participated in the workshops belong to. Where the construction sites are located (geographically). It is important that you provide photographic information of the works in which the workers work when the survey was carried out.

Our answer: Information about participants have been added in line 84 and locations in line 82. Also, for ethical reasons we cannot provide pictures from specific work sites connected to the participants. As anonymity was a part of the agreement prior to interviews.

Table 3 is not correctly framed on the page and it is not known if any columns are missing.

Our answer: This has been corrected.

3.1. Analysis - things that work and things that don't 249

Lines 256-257: "Some explained that the action plan had been canceled, because a similar initiative was already in place. Some explained that they just had not acted on the action plan." Please give a further explanation of this comment. Was the company unaware that this investigation was being done? Did the company managers sign and accept the informed consent?

Our answer:  the managers were not the participants in this workshop programme, or study. Therefore they did not sign informed consent. They did however agree to sending their OSH professionals to the course, so in that sense they have shown commitment. The OSH professionals signed themselves up to participate, so in that sense, it was not the responsibility of the researchers to make sure that managers were informed.

Lines 259-262: "I: 'So the action plan, how did it go with that?' 259

Q: 'Mmh, I mean. We didn't really do as much as I wished. But sometimes, you just don't get something… I mean all the things you want. '"You cannot allow the reader to draw a subjective and random conclusion from this comment between the interviewer and the worker. Please explain what it means for research this comment contributed by the worker.

Our answer:  we agree that some perspective should be provided for this comment and have added the following to the analysis:

This powelessness suggested as a reason for not implementing the participant’s action plan, is in agreement with previous research on OSH professionals. Here it has been described that OSH professionals often have a hard time gaining traction within their organizations, and that often they are encouraged to perform administrative, legitimizing or socially oriented tasks, rather than tasks that actually change the physical safety and health of workers (Provan et al. 2019).

3.1.1 Improved knowledge about physical strain in construction work 294

3.1.2 Applying OSH knowledge in practice 349

3.1.3 The course as a means for network and ability to share knowledge 418

3.1.4 Improving knowledge and awareness of implementation practices 476

What do all the comments raised in all section 3.1 contribute to the investigation?

Our answer:  they provide knowledge and context for understanding what a workshop based course such as the one investigated may lead to. They are a large part of the background for the conclusions in the study. The conclusions and introductions further answer this question.

Please explain the term "(re) configured"; it is used very often in the text.

Our answer:  (re)configured is a theoretical concept derived from agential realism a science of science discipline. That phenomena are (re)configured rather than that ‘truth’ is discovered about them refers to the never-ending changeability of the universe and phenomena in it. When a phenomenon is (re)configured through some form of science, it aquires new characteristics, it can be used for new purposes and/or contribute to new understandings. This (re)configuration is however again never finite and may in new situations e.i. science again be (re)configured, for instance when we acquire new technologies, other concepts or methods for investigation.

We would prefer not to embark on a large science of science discussion explaining this, as that would sort of divert attention away from the point of the article. We do however acknowledge that this concept is not familiar to all scientists and therefore we provided the following in line 196 (given meaning and characteristics both in the understanding of humans and in their worldly functioning (Barad, 2007)).

Please explain how the workshops were carried out, how the workshops were structured, the agenda, questionnaires, tasks, questions. Did the workers have a paper agenda or was it all a PowerPoint presentation? Where were the workshops held, in a company room or in a place set up for such an event?

Our answer:  This information has been described in further detail in section 2.3.

The workshops have been designed for higher category workers (white collar). Explain why the workshops for lower-ranking workers (blue collar) have not been held. It is not understood why the keywords use "blue collar" when the research has been carried out on the highest professional category (white collar). It is recommended to remove the term "blue collar workers" from the keywords or change the name of the keyword to "white collar workers". If not, please explain what were the professional categories of the workers who participated in this research.

Our answer: We see your point. However, as all the OHS-professionals were representing construction companies or served as consultants in the construction industry they used the knowledge from the course in the construction industry. Therefore we believe that the term “white collar workers” would be more confusing. We have specified that the participants all represents blue collar workers in line 90.  

  1. Discussion
  2. Conclusion

Please expand the conclusions. They are very poor for the research that has been done. Surveys with qualitative values and quantitative values have been obtained, so the conclusions can be extended further.

Our answer:  Good point. We have expanded the conclusion.

Indicate what implications this type of workshops can have for the improvement of health and safety conditions in construction companies based on the results obtained.

Our answer:  This information has been added in line 733.

References

Thanks for the modifications. The References section is correct and clear.

Our answer: Thank you

Round 3

Reviewer 2 Report

I suggest that the authors expand the conclusions and future considerations regarding this research; how this type of survey could be adapted to construction sites; Are the current training courses on risk prevention effective?; Do External Prevention Services propose this type of improvement in prevention?; etc. I cannot understand that the results of the investigation serve to write a single paragraph of conclusions.

This manuscript is a resubmission of an earlier submission. The following is a list of the peer review reports and author responses from that submission.

Round 1

Reviewer 1 Report

The paper aims to examine how the participants use and incorporate research-based knowledge from a three-day training course into practice. 

overall the paper is well written, needs minor changes and revisions. 

In abstract section, please define OSH the first time. 

fonts used in tables is different 

Questions used in phone interview maybe listed as a table instead of the paragraph. 

In the results section, the qualitative responses from interview people were very long to read. You may summarize some of the responses with similar ideas. 

Please add a short paragraph about the implications of this study and how it may benefit other industries. 

Reviewer 2 Report

This manuscript describes an evaluation of the effectiveness of a 
"research to practice" effort to determine successful implementation of knowledge gained from training workshops to reduce harms from physical exertion at the workplace. Study participants were presented results from research and new knowledge on how to implement strategies to reduce physical exertion, encouraged to develop action plans for interventions at their work place and then success for each participant was evaluated by surveys and phone interviews.  Overall, participants ranked the knowledge gained and it's value very relevant to their work. Plans to incorporate the training into interventions at their workplace was also highly ranked. Immediately after training and on follow-up actual implementation was not as highly scored but, only 90 days had passed since the training.  The most mentioned barriers to implementation were "lack of time", "organizational changes" and "resistance from leaders"

Overall this manuscript is well written. It could be improved with careful editing by a native speaker of English and could be condensed to a great extent. Specific comments and changes are on the attached manuscript. 

For the results this reviewer suggests that the sequence of the tables  be changed so that Table 4 (Action Plans) is renumbered as Table 2, Table 2 renumbered as Table 3 and Table 3 renumbered as Table 4

Reviewer 3 Report

This manuscript regards and important problem - musculoskeletal disorders. They are still a problem across different working sectors.  Prevention of musculoskeletal pain should be a high priority in jobs characterized by repeated manual lifts, including heavy lifts, and work with bends and twists in the back.

Only 17 participants were examined by phone how they were satisfied with workshop regarding  about physically demanding work.